# Lymphopenia and Early Variation of Lymphocytes to Predict In-Hospital Mortality and Severity in ED Patients with SARS-CoV-2 Infection

**DOI:** 10.3390/jcm11071803

**Published:** 2022-03-24

**Authors:** Maxence Simon, Pierrick Le Borgne, François Lefevbre, Sylvie Chabrier, Lauriane Cipolat, Aline Remillon, Florent Baicry, Pascal Bilbault, Charles-Eric Lavoignet, Laure Abensur Vuillaume

**Affiliations:** 1Emergency Department, University Hospital of Strasbourg, 67000 Strasbourg, France; maxence.simon@chru-strasbourg.fr (M.S.); pierrick.le-borgne@chru-strasbourg.fr (P.L.B.); sylvie.chabrier@chru-strasbourg.fr (S.C.); florent.baicry@chru-strasbourg.fr (F.B.); pascal.bilbault@chru-strasbourg.fr (P.B.); 2Regenerative NanoMedicine (RNM), Fédération de Médecine Translationnelle (FMTS), INSERM (French National Institute of Health and Medical Research), UMR 1260, Strasbourg University, 67000 Strasbourg, France; 3Department of Public Health, University Hospital of Strasbourg, 67000 Strasbourg, France; francois.lefevbre@chru-strasbourg.fr; 4Service d’Accueil des Urgences SAMU 57, CHR Metz-Thionville, 57085 Metz, France; l.cipolat@chr-metz-thionville.fr (L.C.); a.remillon@chr-metz-thionville.fr (A.R.); 5Urgences SAMU SMUR, Hôpital Nord Franche Comté, 90015 Belfort, France; charles-eric.lavoignet@hnfc.fr

**Keywords:** COVID-19, lymphopenia, mortality

## Abstract

(1) Introduction: Multiple studies have demonstrated that lymphocyte count monitoring is a valuable prognostic tool for clinicians during inflammation. The aim of our study was to determine the prognostic value of delta lymphocyte H24 from admission from the emergency department for mortality and severity of SARS-CoV-2 infection. (2) Methods: We have made a retrospective and multicentric study in six major hospitals of northeastern France. The patients were hospitalized and had a confirmed diagnosis of SARS-CoV-2 infection. (3): Results: A total of 1035 patients were included in this study. Factors associated with infection severity were CRP > 100 mg/L (OR: 2.51, CI 95%: (1.40–3.71), *p* < 0.001) and lymphopenia < 800/mm^3^ (OR: 2.15, CI 95%: (1.42–3.27), *p* < 0.001). In multivariate analysis, delta lymphocytes H24 (i.e., the difference between lymphocytes values at H24 and upon admission to the ED) < 135 was one of the most significant biochemical factors associated with mortality (OR: 2.23, CI 95%: (1.23–4.05), *p* = 0.009). The most accurate threshold for delta lymphocytes H24 was 75 to predict severity and 135 for mortality. (4) Conclusion: Delta lymphocytes H24 could be a helpful early screening prognostic biomarker to predict severity and mortality associated with COVID-19.

## 1. Introduction

With more than 5 million deaths worldwide, the coronavirus disease 19 (COVID-19) pandemic continues to place a heavy burden on healthcare systems [1]. From the beginning, triage of patients on arrival to the emergency department (ED) has been a primary focus of resource management [2]. Several prognostic scores have been developed to anticipate the need for intensive care unit (ICU) admission, including analysis of clinical and biological parameters [3,4,5,6]. The use of machine learning-based tools has also been proposed for prognostic assessment and triage [7]. Lymphopenia, defined as a lymphocyte count of less than 1500/mm^3^, is present in the majority of scores and tools [8].

The analysis of lymphocyte count, and more specifically of lymphopenia, has often been studied as a marker in inflammatory and infectious diseases [9,10,11]. Lymphopenia is indeed an already demonstrated prognostic marker in sepsis and cancer [12,13,14]. In viral and inflammatory infections, many pro-inflammatory cytokines, including Interleukin-6, cause metabolic dysfunction and a decrease in lymphocytes proportional to the severity of the infection [15,16,17]. Moreover, the very notion of persistent lymphopenia is also a poor prognostic factor in sepsis [18].

These same observations were quickly made in SARS-CoV-2 infection during the first wave of the outbreak epidemiological studies [19,20,21]. Thus, a lymphocyte count of less than 20% of total leukocytes has been suggested as a marker of poor prognosis [22]. A meta-analysis on nearly 5000 patients evaluated the correlation between the lymphopenia threshold and the severity and mortality with respective ORs of 4.2 CI 95% (3.46–5.09) and 3.71 CI 95% (1.63–8.44) [23]. The use of lymphopenia has also been suggested as a biomarker for the diagnosis of SARS-CoV-2 infection with sensitivities of 72% and 94% respectively for the thresholds of 1100/mm^3^ and 2000/mm^3^ [10].

Lymphopenia is therefore well described as a predictive biomarker of the severity of SARS-CoV-2 infection. Nevertheless, few studies have, to our knowledge, looked at the variability of this biomarker over time, and within the first 24 h after admission. In this context, the main objective of our study was to investigate the prognostic value of the early variation of the lymphocyte count between admission to the ED and H24 (∆Lymphocytes H24) in patients infected with SARS-CoV-2.

## 2. Materials and Methods

### 2.1. Study Population and Settings

The Great East of France has been severely affected by the SARS-CoV-2 pandemic. Thus, we conducted a retrospective multicenter study in six ED of this region (Regional University Hospital of Strasbourg, Regional University Hospital of Reims, Colmar Hospital, Nord Franche-Comteé Hospital, Metz-Thionville Regional Hospital, and Haguenau Hospital).

We included all adult patients who were hospitalized for COVID-19 after presenting to the ED in the first wave of the epidemic in France (between 1 March and 30 April 2020). The inclusion criteria were all patients with a laboratory-confirmed diagnosis of COVID-19 by RT-PCR on a nasopharyngeal swab and hospitalized after admission in ED in the participating centers. The exclusion criteria were patients who received palliative therapy or limitation of therapeutic effort upon admission to the ED, patients with a medical history or treatment that altered their blood cell counts (e.g., chemotherapy, immunosuppressive therapy, oral or inhaled long- and short-term corticosteroid therapy, pre-admission antibiotic therapy, active cancer, or hematological malignancies).

### 2.2. Data Collection

We retrospectively studied patients’ electronic medical records for epidemiological, clinical, and biochemical data and then standardized the results in a report file. We recorded symptom onset date along with patient’s current treatment and medical history (including cardiovascular disease, diabetes, pre-existing renal failure, cancer, and hematological diseases). The primary endpoint was the in-hospital mortality. The secondary endpoint was the severity of the disease. Severe disease was defined by patient admission to the ICU (patients requiring invasive mechanical ventilation), and moderate disease was defined by patient admission to conventional hospitalization units (most patients with oxygen therapy). Overweight was defined by a body mass index superior to 25 kg/m^2^. Functional autonomy was measured by the Knaus score [24]. Standard biochemical parameters, such as levels of creatinine, CRP, total leukocytes, and lymphocytes, were also collected. Lastly, we measured lymphocyte and early (∆Lymphocyte H24), i.e., the difference between lymphocytes values at H-24 and upon admission to the ED.

### 2.3. Statistics

The descriptive statistical analysis of the categorical variables was performed by providing the frequencies and the proportions. For each continuous variable, the median and the first and third quartiles were given. Wilcoxon tests were performed to continuous covariates. To compare the categorical covariates, Chi-Squared tests or Fisher tests were performed. A multivariate logistic model was performed with the statistically significant and clinically relevant covariates (i.e., age, gender, Body mass index, Knaus score, C reactive protein, Creatinine (only for the mortality), admission lymphocytes, lymphocytes H24, ∆lymphocytes. We performed ROC curves to predict the best lymphocyte and ∆Lympho thresholds to predict severity and mortality. The comparisons of AUC were carried out with the DeLong test. Analyses were performed with R software (version 4.0.2) (R Core Team, Vienna, Austria).

## 3. Results

### 3.1. Characteristics of the Study Population

During the study period, a total of 49,326 patients were admitted to the ED of all six hospitals. Of these patients, 4470 (9.1%) had a laboratory-confirmed SARS-CoV-2 infection and, in fine, 1035 patients were included in our study (Flowchart in Appendix A). Our cohort had a median age of 69 (58–79) years and was predominately male (58.9%, CI 95%: (55.8–61.8). Two-thirds of the study population was overweight (69.2%). In terms of medical history, over half of the patients (56.7%) had high blood pressure, over a quarter of them (26.7%) had a history of diabetes, and 23.2% of them presented pre-existing renal failure. The number of missing data for delta lymphocytes 24H was 223 (21.55%). At admission, the median lymphocytes count was significantly lower in the group presenting severe disease compared to the moderate disease group 780/mm^3^ (590–1123) *p* = 0.003) versus 900/mm^3^ (640–1220). Our findings were similar at H-24: 800 mm^3^ (570–1110) versus 1010 mm^3^ (710–1360), *p* < 0.001 (Table 1).

### 3.2. Biochemical Factors Associated with Severe COVID-19

Of the total study population, 789 patients (76.2%) had a moderate disease, whereas 246 (23.8%) had a severe disease which required ICU management. In univariate analysis, the factors associated with the severity of the infection were CRP > 100 mg/L (OR: 2.65, 95% CI: (1.98–3.56), *p* < 0.001), lymphopenia < 800/mm^3^ (OR: 1.70, 95% CI: (1.27–2.27, *p* < 0.001), and negative ∆L-H24 (OR: 2.03, 95% CI: (1.48–2.78); *p* < 0.001). In multivariate analysis, the factors associated with the severity of the infection were CRP > 100 mg/L (OR: 2.51, 95% CI: (1.70–3.71), *p* < 0.001), lymphopenia < 800/mm^3^ (OR: 2.15, 95% CI: (1.42–3.27, *p* < 0.001), and negative ∆L-H24 (OR: 3.16, 95% CI: (2.11–4.75); *p* < 0.001). These values are summarized in Table 2.

### 3.3. Predictive Factors of Severe COVID-19

We determined a ROC curve to predict the risk of disease severity. Regarding lymphocytes value at admission, the area under the curve (AUC) was 0.56 (95% CI: (0.52–0.60), *p* = 0.004). The most efficient cutoff to predict the risk of infection severity was 795 lymphocytes; it yielded a sensitivity of 52.5% (95 % CI (46.0–58.9)) and a specificity of 60.7% (95 % CI (57.1–64.1)). Regarding ∆ lymphocytes H24 value, the area under the curve (AUC) was 0.61 (95% CI: (0.57–0.65), *p* < 0.001). The best cutoff to predict the risk of infection severity was a difference between lymphocytes H24 and lymphocytes admission of 75; it yielded a sensitivity of 68.4% (95 % CI (61.7–74.6) and a specificity of 52.5% (95 % CI (48.4–56.6)). These results are summarized in Figure 1.

The effects of the Lymphocyte count to predict the severity are the same for male and female patients but are more important for the youngest and the oldest patients (Appendix A). The effects of the ∆ lymphocytes H24 to predict the severity is more important for the youngest patients and less important for the oldest (Appendix A).

To predict the severity, the AUC of the ∆ lymphocytes H24 (61.1%) is slightly more important than the AUC of the lymphocyte count (56.2%) but the difference is not statistically significant (*p* = 0.122).

### 3.4. Biochemical Factors Associated with Mortality

In our study cohort, 139 patients died during their hospital stay (13.6%). In univariate analysis the factors associated with mortality were lymphopenia < 800/mm^3^ and <500/mm^3^ with respectively (OR: 2.12, 95% CI: (1.47–3.06), *p* < 0.001) and (OR: 2.22, 95% CI 95: (1.42–3.50), *p* < 0.001) and ∆L-H24 < 135 (OR: 2.32,95% CI 95: (1.45–3.73), *p* < 0.001) (Table 3).

### 3.5. Predictive Factors of Mortality

We determined a ROC curve to predict the risk of death. Regarding lymphocytes value at admission, the area under the curve (AUC) was 0.62 (95% CI 95: (0.57–0.67), *p* < 0.001). The best cutoff to predict the risk of death was 885 lymphocytes; it yielded a sensitivity of 68.9% (95% CI 95 (60.4–76.6)) and a specificity of 52.0% (95% CI 95 (48.6–55.4)). Regarding ∆ lymphocytes H24 value, the area under the curve (AUC) was 0.59 (95 % CI: (0.54–0.64), *p* < 0.001). The best cutoff of ∆ lymphocytes H24 to predict the risk of death was 135; it yielded a sensitivity of 76.4% (95% CI (67.2–84.1)) and a specificity of 41.8% (95% CI (38.1–45.5)). These results are summarized in Figure 2.

To predict the mortality, the AUC of the ∆ lymphocytes H24 (59.1%) is slightly less important than the AUC of the lymphocyte count (62.1%) but the difference is not statistically significant (*p* = 0.584).

The effects of the Lymphocyte count to predict mortality are more important for male than female patients and are more important for the youngest patients (Appendix A). The effects of the ∆ lymphocytes H24 to predict mortality is more important for female than male patients and are more important for the 58, 69 years old patients (Appendix A).

## 4. Discussion

The main objective of our study was to evaluate the prognostic value of early lymphocyte count at ED and at H24, in a cohort of patients infected with SARS-CoV-2. We strictly selected patients in order to limit biases that could alter the blood count. Our study confirmed the value of lymphopenia and highlighted the H24 ∆ lymphocytes as a relevant biomarker in the prognostic evaluation of both severity and mortality in COVID-19.

Our results on lymphopenia alone are similar to other studies previously cited. However, lymphocyte variation is less described. Tan et al. were the first to describe a prognostic model of the severity of SARS-CoV-2 infection based on monitoring the proportion of lymphocytes among leukocytes [22]. Indeed, the 11 patients with severe disease (out of 90) did not have a lymphocyte count higher than 20% of the total leukocyte count or 1500/mm^3^ at follow-up. Similarly, lymphocyte counts below 5% (375/mm^3^) were only present in patients with severe diseases. Chen et al. showed a tendency for lymphopenia to be more marked and long-lasting in the most severe patients [25,26]

Several pathophysiological hypotheses make it possible to link the severe forms to the depth of lymphopenia. First, SARS-CoV-2 induces an aggression mainly of mononuclear cells in part through the ACE2 receptor present on lymphocytes [27,28,29]. Second, hematopoietic precursor damage in COVID-19 could also explain the lymphopenia [30]. Thirdly, lymphocyte metabolism is disturbed by hyperlactatemia frequently encountered in severe forms of infection. Indeed, lactate inhibits in particular the mobility of CD4+ cells and blocks the efficiency of the cytolytic functions of CD8 [31]. Finally, it could be hypothesized that the infection of the adrenal and cerebral parenchyma, by the production of the ACE2 receptor, induces an over-solicitation of the corticotropic axis and then its exhaustion in the severe forms [28,29]. This would explain in part the lymphopenia and eosinopenia caused by the initial hypercorticism, as well as the secondary hypocorticism and the effectiveness of corticosteroids in the most severe forms [32,33]. The kinetics of lymphopenia is then important to take into account since it is directly correlated to its severity.

COVID-19 patients require ICU care more frequently than during the influenza infection [34]. The direct consequence of this need for ICU is the shortage of resources such as ventilators, which can lead to an increase in overall morbidity and mortality [35,36]. The medical decision to introduce or stop artificial ventilation is not based solely on medical considerations. Our cohort, from the first wave, includes many patients on artificial ventilation, well before the emergence of corticoids, vaccines, and high flow non-invasive ventilation tools [37]. Helping to predict the severity of the disease, using adapted assessment tools and scores, would be closer to reality [38]. For example, the AIFELL score was developed to estimate the risk of hospitalization. It includes 1 point granted for lymphopenia below the threshold of 1450/mm^3^ when a score greater than 4 warrants hospitalization [39]. The COMPASS-COVID-19 score uses multiple parameters including lymphocyte values and has an area under the curve of 0.77 (Se 81%, Spe 60%) [4]. A hematocytometric score analyzing blood counts over 3 days has an AUC of 0.753 (95% CI 0.723–0.781) and increases to 0.875 (95% CI 0.806–0.926) by day 3 [5]. Finally, Xie et al. developed a clinical and biological prognostic score which includes lymphocytes with internal and external validities at 0.89 and 0.98 [6]. Thus, the use of early lymphocyte variation between admission to the emergency room and H24, by allowing time for observation in hospital before decision-making, could be fully integrated into dynamic scores to include a notion of evolution with time.

### Limitations

However, our study has some limitations. Firstly, its retrospective nature implies inherent biases, although reduced by exclusion criteria (including pathologies that can modify the leukocyte formula). Secondly, the severity of the pandemic, which particularly affected the Greater-East region at that time, put great pressure on the health system. We may therefore have underestimated the number of patients with moderate disease as they did not seek care in the ED. This element is also observed in the relative pre-existing autonomy of severe patients in our results, the latter having been selected. Thirdly, our study was conducted in the first wave, when the alpha variant was in the majority. It is possible that these findings will vary with the other variants. Finally, the initial management guidelines included broad indications for mechanical ventilation. This impacted the final cohort as it included many intubated and deceased patients [40].

## 5. Conclusions

The lymphocyte count and its early variation at H24 could be useful in the prognostic evaluation and triage of SARS-CoV-2-infected patients when fully integrated into triage scores and tools.

## Figures and Tables

**Figure 1 jcm-11-01803-f001:**
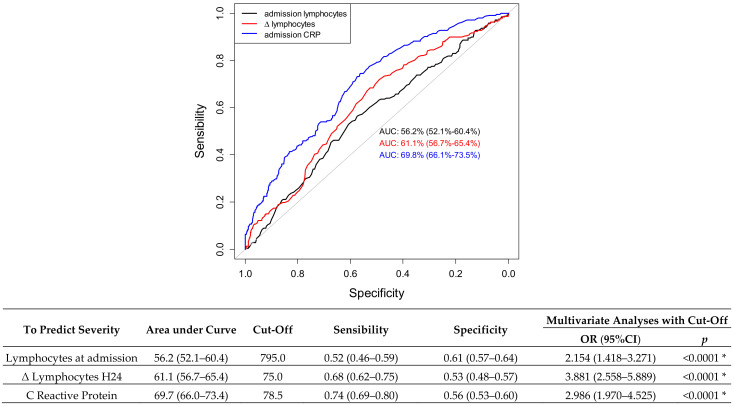
Receiver operating characteristics curve (ROC) for the ability of lymphocyte and early version to predict severity. Legend: CRP = C reactive protein, * = *p* < 0.005.

**Figure 2 jcm-11-01803-f002:**
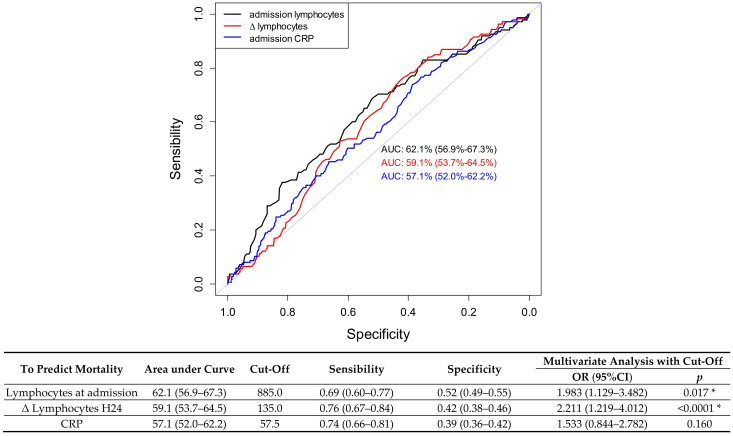
Receiver operating characteristics curve (ROC) for ability of lymphocyte and early variation to predict mortality. Legend: CRP = C reactive protein, * = *p* < 0.005.

**Table 1 jcm-11-01803-t001:** Demographic, baseline, and laboratory characteristics of patients with COVID-19.

	All Patients *n* = 1035	Moderate *n* = 789	Severe *n* = 246	*p*
General characteristics				
Age (years)	69 (58–79)	70 (58–81)	66 (57–72)	<0.001 *
Gender male	609 (58.8)	433 (54.9)	179 (71.5)	<0.001 *
Overweight (BMI > 25)	527 (69.2)	370 (67.0)	157 (74.8)	0.039 *
Chronic medical illness				
hypertension	587 (56.7)	453 (57.4)	134 (54.5)	0.416
Diabetes mellitus	275 (26.7)	202 (25.6)	73 (26.6)	0.207
CKD	237 (23.2)	199 (25.5)	38 (15.8)	0.002 *
Cardiovascular illness	357 (34.5)	291 (36.9)	66 (26.8)	0.004 *
Total autonomy	796 (77.2)	569 (72.4)	227 (92.7)	<0.001 *
Laboratory Findings				
CRP > 100 mg/L	418 (40.7)	275 (35.0)	143 (58.9)	<0.001 *
Lymphocytes (/mm^3^)	870 (630–1200)	900 (640–1220)	780 (590–1123)	0.003 *
Lymphocytes H24 (/mm^3^)	940 (670–1300)	1010 (710–1360)	800 (570–1110)	<0.001 *
Lymphocytes < 800/mm^3^	433 (42.5)	305 (39.4)	128 (52.5)	<0.001 *
∆ lymphocytes H24	60 (−123–290)	100 (−100–323)	−20 (−153–140)	<0.001 *
Outcome				
Hospital stay (days)	10.0 (7.0–17.3)	8.0 (6.0–12.0)	24.0 (17.0–38.0)	<0.001 *
Intra-hospital mortality	139 (13.6)	82 (10.4)	57 (24.1)	<0.001 *

Data are expressed in median (Q1–Q3) or *n* (%), where *n* is the total number of patients with availble data. * *p* < 0.05. Abbreviations = BMI: body mass index, CKD = chronic kidney disease, CRP = C reactive protein, ∆lymphocytes H24 = difference between lymphocytes H24—lymphocytes admission.

**Table 2 jcm-11-01803-t002:** Biochemical factors associated with severe COVID-19 (admission to the ICU).

				Multivariate Analysis **	
	All	Moderate	Severe	OR (95% CI)	*p*-Value
CRP > 100 mg/L	418 (40.7)	275 (35.0)	143 (58.9)	2.51 (1.40–3.71)	<0.001 *
Lymphopenia < 800/mm^3^	433 (42.5)	305 (39.4)	128 (52.5)	2.15 (1.42–3.27)	<0.001 *
∆ lymphocytes H24 < 0	322 (39.7)	211 (35.2)	111 (52.4)	3.16 (2.11–4.75)	<0.001 *

Data are expressed in median (Q1–Q3) or *n* (%), where *n* is the total number of patients with available data. * *p* < 0.05, ** model adjusted for age, gender, body mass index, C reactive protein, creatinine, admission lymphocytes, lymphocytes H24, ∆lymphocytes. Abbreviations: OR = odds ratio, CRP = C reactive protein, ∆ lymphocytes H24 = difference between lymphocytes H24—lymphocytes admission.

**Table 3 jcm-11-01803-t003:** Biochemical factors associated with mortality of COVID-19.

				Multivariate Analysis **	
	All	Survivors	Non-Survivors	OR (95% CI)	*p*-Value
Age > 75 years	328 (32.1)	247 (27.9)	81 (58.3)	2.74 (1.57–4.80)	<0.001 *
CRP > 100 mg/L	412 (40.5)	345 (39.2)	67 (48.9)	1.48 (0.87–2.52)	0.151
Lymphopenia (<800/mm^3^)	428 (42.5)	349 (40.0)	79 (58.5)	1.88 (1.09–3.25)	0.023 *
∆ lymph H24 < 135	487 (60.7)	406 (58.3)	81 (76.4)	2.23 (1.23–4.05)	0.009 *

Data are expressed in median (Q1–Q3) or *n* (%), where *n* is the total number of patients with available data. * *p* < 0.05, ** model adjusted for age, gender, body mass index, C reactive protein, creatinine, admission lymphocytes, lymphocytes H24, ∆lymphocytes. Abbreviations: OR = odds ratio, CRP = C reactive protein, ∆ lymphocytes H24 = difference between lymphocytes H24—lymphocytes admission.

## Data Availability

The data presented in this study are available on request from the corresponding author.

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
