# Peer review of "Lymphopenia and Early Variation of Lymphocytes to Predict In-Hospital Mortality and Severity in ED Patients with SARS-CoV-2 Infection"

_jcm, 2022, doi:10.3390/jcm11071803_

Round 1
Reviewer 1 Report
Multiple studies have demonstrated that lymphopenia is a valuable prognostic biomarker predicting worse clinical outcomes in COVID-19. The study by Simon et al. further characterizes the prognostic value of delta lymphocyte H24 for mortality and severity of COVID-19. The data are presented clearly and the authors' conclusions appear appropriate and are supported by the data.
I have a couple of comments:
- Older age is associated with decreasing lymphocyte counts. It would be very interesting to know can the prognostic value of lymphopenia or Δ lymphocytes H24 on the outcome of COVID 19 patients be influenced by age or gender? Please, can the authors analyze with similar approaches in patients stratified by age groups or gender?
- The analysis comparing lymphopenia verse Δ lymphocytes H24 for predicting worse clinical outcomes of COVID-19 is not described, although both were shown to be significant predictors. Lymphopenia or Δ lymphocytes H24, which one is superior predictor or both are similar to predict worse outcomes in COVID-19 patients? Can the authors give some comparison analysis of the AUCs between the two predictors and comment on this in the discussion?
- Page 3 rows 103-104: The authors state a multivariate logistic model was performed with the statistically significant and clinically relevant covariates. Can the authors describe the relevant covariates included in the logistic models in Statistics (page 3)?
Author Response
#Reviewer 1
Multiple studies have demonstrated that lymphopenia is a valuable prognostic biomarker predicting worse clinical outcomes in COVID-19. The study by Simon et al. further characterizes the prognostic value of delta lymphocyte H24 for mortality and severity of COVID-19. The data are presented clearly and the authors' conclusions appear appropriate and are supported by the data.
Response: We thank the reviewer for his comments and hope that the revised version will meet his expectations.
I have a couple of comments:
- Older age is associated with decreasing lymphocyte counts. It would be very interesting to know can the prognostic value of lymphopenia or Δ lymphocytes H24 on the outcome of COVID 19 patients be influenced by age or gender? Please, can the authors analyze with similar approaches in patients stratified by age groups or gender?
Response: We thank the reviewer for this comment. We have added these elements in our results and the data set in additional material ( not to overload the manuscript)
- The analysis comparing lymphopenia verse Δ lymphocytes H24 for predicting worse clinical outcomes of COVID-19 is not described, although both were shown to be significant predictors. Lymphopenia or Δ lymphocytes H24, which one is superior predictor or both are similar to predict worse outcomes in COVID-19 patients? Can the authors give some comparison analysis of the AUCs between the two predictors and comment on this in the discussion?
Response: To predict the severity, the AUC of the ∆ lymphocytes H24 is slightly more important than the AUC of the lymphocyte count but the difference is not statistically significant (p=0.122). To predict the mortality, the AUC of the ∆ lymphocytes H24 is slightly less important than the AUC of the lymphocyte count but the difference is not statistically significant (p=0.584). In view of these elements it did not seem important to us to discuss them.
- Page 3 rows 103-104: The authors state a multivariate logistic model was performed with the statistically significant and clinically relevant covariates. Can the authors describe the relevant covariates included in the logistic models in Statistics (page 3)?
Response: The statistically significant and clinically relevant covariates are age, gender, Body mass index, Knaus score, C reactive protein, Creatinine (only for the mortality), admission lymphocytes, lymphocytes H24, ∆lymphocytes. We have specified these elements in the manuscript.
Reviewer 2 Report
The authors present an interesting manuscript regarding the potential clinical value of difference in lymphocyte counts between 24 hours and admission (delta lymphocytes 24H) as severity and mortality in the early covid-19 pandemic. They claim that low delta lymphocytes 24H values are potential predictors for severity and mortality. In general, the paper is very well-written and has a clear line, the statistical methods seem sound and relevant. However, I have some problems with the consistency of the presented numeric data presented, it may be due to my own lack of understanding or due to inaccuracy from the authors.
Abstract. Delta lymphocytes 24H should be defined already in the abstract.
Materials and Methods:
“Patients with a medical history or treatment that altered their blood counts and, therefore, their circulating lymphocytes or neutrophils (e.g., chemotherapy, immunosuppressive therapy, long- and short-term corticosteroid therapy, pre-admission antibiotic therapy, active cancer, or hematological malignancies) were also excluded from our study.”
How was chemotherapy, immunosuppressive therapy defined?
Were also low dosages of corticosteroid therapy excluded? What about inhaled corticosteroids?
Results:
What were the numbers of missing data, in particular regarding delta lymphocytes 24H?
It is stated that:
Thus, the variation in lymphocytes was more significant in the group of patients admitted to intensive care with a respective delta of 335 (30–700) for the moderate group and 760 (243–1238) for the severe group. (Table 1).
However, I cannot find these values in table 1. The described median delta lymphocytes for the moderate and severe groups are 100 and -20 respectively. Furthermore, for the severe group, the difference from 800 (24H) to 780 (admission) is 20, not -20?
Next, there seem to be an inconsistency between table 1 and 2. The CRP and lymphopenia values are the same in both tables, but delta lymphocytes H24 are very different.
Table 1:
How do the authors explain why patients in the severe group had a larger fraction of patients with total autonomy. May be a few sentences in the discussion would be appropriate?
Figure 1 and 2. Prediction of severity and mortality. ROC analysis, AUC curves. I acknowledge that the authors present the data and the modest sensitivity poor sensitivity. I suggest putting in CRP as well for comparison in both figures.
Discussion:
The discussion is short, I suggest the authors add some reflections regarding the fact that the observations are collected during the initial, alpha dominant, phase of the covid-19 pandemic and my not be relevant for the delta or omicron variants.
I also would like to read a brief reflection regarding the possible reasons/mechanism for the severity and poor outcome associated with low H24 lymphocyte values.
Author Response
#Reviewer2
The authors present an interesting manuscript regarding the potential clinical value of difference in lymphocyte counts between 24 hours and admission (delta lymphocytes 24H) as severity and mortality in the early covid-19 pandemic. They claim that low delta lymphocytes 24H values are potential predictors for severity and mortality. In general, the paper is very well-written and has a clear line, the statistical methods seem sound and relevant. However, I have some problems with the consistency of the presented numeric data presented, it may be due to my own lack of understanding or due to inaccuracy from the authors.
Response: We thank the reviewer for his comments and hope that the revised version will meet his expectations.
Abstract. Delta lymphocytes 24H should be defined already in the abstract.
Response: We thank the reviewer for his comment and have made the requested correction.
Materials and Methods:
“Patients with a medical history or treatment that altered their blood counts and, therefore, their circulating lymphocytes or neutrophils (e.g., chemotherapy, immunosuppressive therapy, long- and short-term corticosteroid therapy, pre-admission antibiotic therapy, active cancer, or hematological malignancies) were also excluded from our study.”
How was chemotherapy, immunosuppressive therapy defined?
Response: There was no definition beyond the patient's knowledge of these treatments. We made sure by studying the medical records that all the patients included in this study did not have any. Thus, we were able to obtain a cohort as "naive" as possible with respect to SARS-CoV-2 viral infection to study the lymphocyte count. These elements are explained in the methods section. However, if you do not find them clear enough, we can add more details.
Were also low dosages of corticosteroid therapy excluded? What about inhaled corticosteroids?
Response: Yes, all patients receiving long-term oral or inhaled corticoids were excluded from the study. These elements can modify the lymphocyte count.
Results:
What were the numbers of missing data, in particular regarding delta lymphocytes 24H?
Response: The number of missing data for delta lymphocytes 24H is 223 (21.55%).
It is stated that:
Thus, the variation in lymphocytes was more significant in the group of patients admitted to intensive care with a respective delta of 335 (30–700) for the moderate group and 760 (243–1238) for the severe group. (Table 1).
However, I cannot find these values in table 1. The described median delta lymphocytes for the moderate and severe groups are 100 and -20 respectively. Furthermore, for the severe group, the difference from 800 (24H) to 780 (admission) is 20, not -20?
Next, there seem to be an inconsistency between table 1 and 2. The CRP and lymphopenia values are the same in both tables, but delta lymphocytes H24 are very different.
Response: The sentence about the variation in lymphocytes of 335 (30–700) for the moderate group and 760 (243–1238) for the severe group is a typo and have been corrected.
For the severe group, there is a decrease between H24 and the admission so the value is -20. The difference from 800 (24H) to 780 (admission) is 20 but not take into account two factors: the sample is not the same due to missing data and the position parameter is the median, not the mean.
Finally, there is an error in Table 2 that has been corrected in the manuscript. This is the number and proportion of patients with a negative delta.
Table 1:
How do the authors explain why patients in the severe group had a larger fraction of patients with total autonomy. May be a few sentences in the discussion would be appropriate?
Response: Indeed, our study excluded patients who had received a limitation of active therapies from the emergency room. Thus, the subgroup of patients admitted to the ICU is globally autonomous, whereas the other subgroup (less severe) is more heterogeneous and contains patients of variable autonomy, which explains this difference.
In fact, given the pressure of admission to the ICU in our region during the first wave, the patients admitted at that time for mechanical ventilation (no other non-invasive alternative) were somewhat selected among the least fragile patients. We have added a sentence about this in the limitation section to clarify.
Figure 1 and 2. Prediction of severity and mortality. ROC analysis, AUC curves. I acknowledge that the authors present the data and the modest sensitivity poor sensitivity. I suggest putting in CRP as well for comparison in both figures.
Response: This was added in both figures.
Discussion:
The discussion is short, I suggest the authors add some reflections regarding the fact that the observations are collected during the initial, alpha dominant, phase of the covid-19 pandemic and my not be relevant for the delta or omicron variants.
Response : Indeed, it is important to specify this element. We have added it in the limit section.
I also would like to read a brief reflection regarding the possible reasons/mechanism for the severity and poor outcome associated with low H24 lymphocyte values.
Response: We thank the reviewer for this pertinent idea. On his advice we have added a paragraph. We hope that this new version will meet his expectations.
Round 2
Reviewer 1 Report
The concerns have been addressed.
Reviewer 2 Report
The authors have responded and revised the manuscript sufficiently and well acceptable for publication.